# Enabling Seamless Connectivity: Networking Innovations in Wireless Sensor Networks for Industrial Application

**DOI:** 10.3390/s24154881

**Published:** 2024-07-27

**Authors:** Shathya Duobiene, Rimantas Simniškis, Gediminas Račiukaitis

**Affiliations:** 1Department of Laser Technologies, FTMC—Center for Physical Sciences and Technology, Savanoriu Ave. 231, LT-02300 Vilnius, Lithuania; g.raciukaitis@ftmc.lt; 2Department of Physical Technologies, FTMC—Center for Physical Sciences and Technology, Sauletekio Al. 3, LT-10257 Vilnius, Lithuania

**Keywords:** Internet of Things, wireless sensor network, SSAIL technology, 6LoWPAN, MQTT-SN, novelty detection system, COOJA simulator, web server

## Abstract

The wide-ranging applications of the Internet of Things (IoT) show that it has the potential to revolutionise industry, improve daily life, and overcome global challenges. This study aims to evaluate the performance scalability of mature industrial wireless sensor networks (IWSNs). A new classification approach for IoT in the industrial sector is proposed based on multiple factors and we introduce the integration of 6LoWPAN (IPv6 over low-power wireless personal area networks), message queuing telemetry transport for sensor networks (MQTT-SN), and ContikiMAC protocols for sensor nodes in an industrial IoT system to improve energy-efficient connectivity. The Contiki COOJA WSN simulator was applied to model and simulate the performance of the protocols in two static and moving scenarios and evaluate the proposed novelty detection system (NDS) for network intrusions in order to identify certain events in real time for realistic dataset analysis. The simulation results show that our method is an essential measure in determining the number of transmissions required to achieve a certain reliability target in an IWSNs. Despite the growing demand for low-power operation, deterministic communication, and end-to-end reliability, our methodology of an innovative sensor design using selective surface activation induced by laser (SSAIL) technology was developed and deployed in the FTMC premises to demonstrate its long-term functionality and reliability. The proposed framework was experimentally validated and tested through simulations to demonstrate the applicability and suitability of the proposed approach. The energy efficiency in the optimised WSN was increased by 50%, battery life was extended by 350%, duplicated packets were reduced by 80%, data collisions were reduced by 80%, and it was shown that the proposed methodology and tools could be used effectively in the development of telemetry node networks in new industrial projects in order to detect events and breaches in IoT networks accurately. The energy consumption of the developed sensor nodes was measured. Overall, this study performed a comprehensive assessment of the challenges of industrial processes, such as the reliability and stability of telemetry channels, the energy efficiency of autonomous nodes, and the minimisation of duplicate information transmission in IWSNs.

## 1. Introduction

As the foundation of data collection from different environments and data processing on Internet servers, wireless sensor networks (WSNs) are intended as a framework for Internet of Things (IoT) technology [1]. WSNs are used in the military, public sector, agriculture, surveillance, household, industry, and even on the human body, e.g., the so-called body sensor networks (BSNs). For example, WSNs are used in the military sector for battlefield surveillance and monitoring, for precision farming and crop monitoring in agriculture [2], for home automation and security in smart house applications, and for asset tracking and predictive maintenance in the industrial sector [3]. The economic feasibility of using this network type in numerous applications has led to a surge in interest over the last decade. The cost-effectiveness of WSNs and IoT is influenced by factors such as the low cost and ease of installation of sensor networks (SNs) in industry and consumer infrastructure, energy efficiency, and the value and payback of the data collected and used. 

The proposed WSN investigations fall within the scope of the IoT trend and reach the 3rd and 4th levels (“Cyber” and “Cognition”) in industrial IoT (IIoT) applications [3,4], which are revolutionising the sector, offering opportunities such as predictive maintenance, asset tracking, and intelligent manufacturing processes. In some cases, security and privacy measures [5,6] are also essential features of WSN applications.

The integration of WSNs with IoT technologies brings significant advances in various sectors. The new approach to developing WSNs based on selective surface activation induced by laser (SSAIL) technology, which enables the development of miniaturised sensors with integrated energy-harvesting circuits and antenna circuits, as well as the ability to fabricate three-dimensional circuits on flexible materials, ensures long-term, cost-effective, and sustainable operation, even in harsh conditions. This approach improves the energy efficiency and performance of the device and has been practically implemented in the monitoring of manufacturing processes [7], where embedded temperature and humidity sensor nodes are installed. The rapid development of IoT devices has enabled the implementation of embedded web servers that enhance the capabilities of IWSNs for various applications. These include overload conditions, maintenance alerts, and maintaining the tension of conveyor belts in production lines to ensure smooth operation, quality control, and safety. The proposed customisable sensor systems provide a practical solution for large-scale environmental monitoring. By analysing SN characteristics, this study provides a framework for reducing deployment costs and improving the operational efficiency of WSNs. Overall, this research contributes to advancing WSN and IoT technologies by providing innovative solutions for real-time environmental monitoring and ensuring the reliable and long-term operation of SNs in diverse applications. These advances will significantly benefit cost efficiency, sustainability, and operational efficiency. Our process of design and network innovation in developing a seamless WSN for industrial applications can be used to outline the main contributions of our manuscript:The development of an innovative sensor design using the SSAIL technology to enable the miniaturisation and efficient fabrication of sensors on various plastic bodies to monitor environmental parameters.The presentation of a new approach for the integration of WSN with 6LoWPAN and MQTT-SN in industrial IoT systems, providing innovative solutions for early quantitative assessment and practical use in industrial environments.The integration of the ContikiMAC protocol and its use to improve energy-efficient connectivity, which is essential for the lifetime and efficiency of industrial IoT applications.The implementation of a novelty detection system (NDS), which is an algorithm in sensor nodes used to detect unusual patterns or anomalies in the collected data.The application of efficient particle swam optimisation for the duty cycle (PSO-DC) routing algorithm and its flowchart to optimise the duty cycle of the sensor nodes to improve energy efficiency and prolong the lifetime of the SN.The proposal for a schematic representation of WSN deployment in real time and its initial planning and visualisation on conveyor belts.The simulation model of the network topology for node placement using the COOJA simulator and the analysis of the mobility model of the designed sensor node deployed in a conveyor belt system.The investigation of static and mobile conveyor belt scenarios; the analysis of data transmission latency and the average number of hops of networks after WSN optimisation based on four quadratic domains; the analysis of the number of packets received per node; the calculation of the average duty cycle; and the comparison of the results with those of different optimisation techniques.The estimation of the average power consumption of nodes in different operating modes and the comparison of the results of the optimisation of WSN components.The initial experiments with the developed sensor nodes in the FTMC Department of Laser Technologies to monitor the temperature and humidity parameters in the environment and the power consumption of the communication module are presented.

Overall, these contributions advance the state of the art in WSNs and their applications, providing robust, energy-efficient, and scalable solutions for industrial IoT systems. The article is organised as follows: Section 2 provides an overview of the relevant literature. Section 3 describes the proposed method for integrating 6LoWPAN nodes with MQTT-SN into an adaptive deployment and clustering system for industrial IoT. In Section 4, we report the results, evaluate the performance, and investigate the power consumption. Section 5 discusses future work. The conclusion of this study can be found in Section 6.

## 2. Recent Trends and Progress of WSN in IoT

A targeted database search revealed that only a handful of articles deal with WSN-oriented industrial applications in conveyor system environments. The particular focus of our research on industrial applications sets us apart from other scholars and provides a unique perspective on using WSNs in the real world. The analysis of relevant studies provides context and highlights the progress and limitations in this area. 

Alghofaili et al. present a framework for analysing the impact of WSNs and provide a structured toolkit for evaluating WSN deployments. However, the framework may require an extensive initial setup and is less effective in dynamic environments [1]. Atalla et al. focus on IoT-enabled precision agriculture and promote optimised crop management and resource utilisation. However, they conclude that implementation costs can be high and require technical expertise [2]. Antonini et al. propose an unsupervised anomaly detection system in industrial environments to provide a robust detection system suitable for harsh conditions [3]. They note that the initial setup and training are complex and time-consuming.

Raza et al. provide a comprehensive overview of the current status and future trends in IWSNs [4]. Industry 4.0 relates to coverage approaches, deployment issues, sensing models, and research obstacles. It encompasses numerous WSNs, including SNs, fog, edge computing, distributed control systems, digital twins, and cyber–physical systems. Our method is eligible and flexible enough to efficiently facilitate the rapid deployment of IoT devices on assembly lines to monitor the production process. 

A novel randomised preservation method improves privacy and extends the lifetime of WSNs. It addresses specific challenges related to radio range and its impact on performance and provides practical solutions that can be applied in real-world IoT scenarios [5]. However, its implementation can be complex and requires significant computing resources, impacting battery life. In [6], a comprehensive survey of privacy protection techniques in WSNs was proposed, identifying the main challenges and proposing potential solutions focusing on security and privacy. This approach is primarily theoretical, with few examples of practical implementation, and some of the proposed solutions may not be feasible for all types of WSN deployments. Substantial progress has been made in the field of WSNs as the need for efficient, scalable and reliable monitoring systems has increased.

In our previous research, we developed a method to design a WSN for environmental monitoring by fabricating some sensor components using SSAIL technology [7]. Key contributions included the design and evaluation of a fractal antenna, the optimisation of sensor node placement, the definition of node architecture, the evaluation of power consumption, the simulation of improved low-energy adaptive clustering hierarchy (LEACH) protocols, the estimation of daily energy consumption, and the deployment of real-time data visualisation applications [7]. This comprehensive approach addresses the need for efficient real-time IoT monitoring solutions by integrating different technological layers, such as the SN protocol stack, the modification of communication module hardware for radio frequency (RF) harvesting, efficient routing algorithms, and an asynchronous web server. Notably, studies have emphasised the significant role of energy-efficient protocols and sleep-scheduling mechanisms in improving the longevity of WSNs in off-premises applications. These techniques are characterised by low implementation costs, fast scalability and applicability in production lines and outdoor environments. They provide a practical approach to environmental monitoring with advanced SNs.

The study by Prisantama et al. facilitates the integration of 6LoWPAN into the existing IPv6 infrastructure, improves the interoperability and scalability of IoT networks, and supports low-power and low-cost communication suitable for various IoT applications [8]. However, tunnelling protocols can introduce additional latency and require advanced network configuration and management. Shelby et al. provide detailed knowledge of 6LoWPAN architecture, protocols, and the applications of the latest developments and trends [9]. Adaptive frameworks adapt to changing network conditions and improve reliability. The authors [10] proposed a trust-based framework to increase security in 6LoWPAN networks and reduce costs. The method we propose reduces the complexity of implementation by providing significant expertise and trust-based systems for computational IoT networks. 

The goal of 6LoWPAN is to facilitate Internet connectivity by sending and receiving IPv6 packets over low-power wireless networks. When 6LoWPAN routes data to and from the access point (AP) [11] and draws the network architecture [12,13,14], it implements routing low-power and lossy networks (RPL). RPL is a routing protocol developed for low-power and lossy networks, such as WSNs. This system uses the advanced encryption standard (AES) 128 Link Layer Security, which encrypts data in 128-bit blocks and uses block encryption with a key size of 128/192/256 bits. Although AES is subject to a global standard, it is considered redundant for this type of network and its embedded devices due to the increased power and energy consumption involved in encryption and decryption [15]. This protocol is well suited for the ESP32 family of devices as it operates in sleep mode, which saves battery energy, reduces memory requirements, and bypasses overhead. Recent developments in these networks and sensors influenced sensor prices due to better performance. Specifications include IEEE 802.15.4 in the 2.4 GHz band, an outdoor range of around 200 metres, a data rate of 200 kbps, and a maximum of 100 nodes. Due to their practicality and affordability, low-cost sensors are used in various applications, e.g., in monitoring temperature, humidity, rain, air, pedestrians [16], lighting, etc. 

The 6LoWPAN protocol offers fewer security features than Zigbee and is less robust than Bluetooth and WiFi. It only works at short ranges when the mesh topology is not used. For this reason, a handful of researchers have proposed advanced encryption algorithms such as Speck128, SIMON, tiny encryption algorithm (TEA), and FlexenTech. These provide robust security and are easy to implement. We refer to these algorithms as ‘lightweight encryption methods’. This protocol fits our idea regarding the ESP32 communication module, which supports the further improvement of the algorithms and is very useful in industrial applications.

Gulec et al. propose an energy-efficient distributed routing algorithm tailored to nanosensor networks. It addresses the unique challenges that contribute to progress in this emerging field and offers practical solutions to improve energy efficiency in nanosensor networks, although the implementation of the proposed algorithm [17] can be complex and requires specialised knowledge, validation, and testing to assess its practical applicability. Th use of the bat optimisation algorithm [18] to improve the localisation of sensor nodes in WSNs strikes a balance between exploration and exploitation and provides a novel approach to solving localisation problems with potential applications. However, the complexity of the bat optimisation algorithm may increase computational requirements and affect energy consumption.

With the growing demand for low-cost WSN nodes [8] for future applications, these devices need to operate over an extended period of time. During data collection and transmission in WSN nodes, minimal data protection is required until reaching the base station (BS). Due to their critical importance, various techniques are used to improve the power consumption and security of WSN nodes simultaneously. One such approach is ‘clustering’ [19,20], in which a cluster of neighbouring WSN nodes determines a primary WSN node, called the cluster head (CH), which facilitates data transmission to the BS. This method optimises node communication by using short distances and robust signal transmission, thus reducing the power consumption caused by weak signals during communication. We use the clustering technique when utilising sensor nodes during real-time deployment. In addition, studies have used energy-efficient algorithms, mobility models, and optimisation algorithms to improve the clustering performance, efficiency, and quality of service (QoS) of WSN networks [12,13]. These criteria include the distance between the nodes, the radio signal strength indicator (RSSI), and the power level of the nodes. In addition, they automate the procedure for clustering and security complexity based on the state of the nodes.

One of the key advantages of message queuing telemetry transport for sensor networks (MQTT-SN) is the efficient use of the user datagram protocol (UDP) as a transport protocol [21]. This eliminates the need for a permanent connection, which reduces overheads and increases efficiency. In MQTT-SN, the connection message is divided into three parts, with two optional parts used for the will message. Topic identifiers are used instead of topic names, and predefined topics are part of the specification. Clients can find the gateway via the discovery process, and both the will topic and the messages can be changed during the session.

One of the main reasons for excessive power consumption in wireless communication is idle listening. ContikiMAC mitigates this problem by coordinating the wake-up times of all nodes. To reduce the impact of interference and collisions in wireless channels, ContikiMAC utilises packet tracking techniques [22]. The reliability and efficiency of communication under noisy conditions can be improved with ContikiMAC by stretching packet transmissions. Due to its adaptive behaviour, ContikiMAC can change its duty cycles depending on traffic patterns and network conditions. The medium access control (MAC) layer of the OSI model is the area where ContikiMAC shines. Sensor nodes with limited resources provide mechanisms for appropriate channel access, collision avoidance, and energy-efficient operation. ContikiMAC provides WSNs with energy-efficient communication capabilities and is an integral part of the Contiki OS [23]. Even though the progress in WSNs and their integration into IoT systems is impressive, there are still some challenges. Communication reliability, a major concern, has been addressed by introducing protocols such as ContikiMAC and improved versions of the LEACH protocol. These not only minimise energy consumption but also provide robust communication links.

The analysis of the different lightweight encryption methods has shown that AES consumes a lot of energy. Other publications have proposed lightweight authentication methods for WSNs, and watermarking technology has been implemented in authentication methods to reduce security costs further and extend the battery life of autonomous WSNs. These practical solutions have been developed to address the specific challenges of WSNs and ensure their relevance and effectiveness in real-world scenarios. Lightweight communication protocols like MQTT-SN have shown promise in ensuring reliable data transmission in low-power and low-bandwidth scenarios. The integration of WSNs into IoT frameworks has expanded their applicability, enabling real-time monitoring and control for various applications. This integration is possible by combining advanced technologies such as edge computing with IoT best practices and artificial intelligence (AI) tools to ensure reliable and secure performances. This integration enables on-site data processing, facilitates instant decision making and monitoring, reduces adaptation latency, and improves system responsiveness. IoT sensors collect various types of data from the conveyor belts, while AI systems analyse the information to gain insights.

Despite focusing on the earlier phase of research work in many areas to address the unique challenges of WSNs, including the need for robust security and longer battery life, we have developed and proposed a state-of-the-art lightweight algorithm. The concept of trust between nodes can be utilised to streamline communication processes and reduce costs while ensuring network security [10]. To build trust, the interactions of each node with its neighbours should be evaluated by monitoring elements such as the communication and behaviour of the nodes. Our method provides a lightweight trustworthiness enhancement algorithm that can improve the performance and security of WSNs, offering a promising future to this field. We comprehensively investigated the benefits and limitations of the following studies: structured impact analysis but high initial effort [1]; optimised farming but high cost and expertise required [2]; and effective anomaly detection but complex setup [3]. It is crucial to consider the specifics of the WSN deployment scenario when determining how to build propagation models, also known as path loss models. Therefore, the following measures are common practice:−To make necessary adjustments to the validated models;−To create a new model to characterise the attenuation better;−To proceed with the application of the updated models.

Our research differs by developing an innovative sensor design using the SSAIL technology and introducing a pioneering approach. The ‘novelty detection system’ (NDS) consolidates frameworks and toolkits for the impact analysis of WSNs. Precision agriculture, industry, forestry, and weather monitoring systems can all benefit from the proposed method. These methods are easier to implement and consume less energy than traditional algorithms, such as AES and PSO. In addition, lightweight authentication methods and surface-modified sensor nodes, altered with the SSAIL technology, are proposed to improve security further and reduce energy consumption. With the increasing demand for low-cost WSN nodes that can operate for long periods, it is crucial to implement techniques that improve both power consumption and security. This innovative technology ensures data integrity, timeliness, and authentication. 

Clustering, in which a CH facilitates data transmission to the BS, optimises communication by reducing the power consumption for signal transmission over short distances. Therefore, clustering was used in combination with lightweight encryption methods for our application to ensure efficient and secure data transmission while minimising energy consumption. This article then looks at the application of this flexible and unsupervised NDS to monitor conveyor belt production under difficult operating conditions in terms of energy, communication, physical access, and security. The NDS revolutionises the concept of trust in WSN nodes by increasing security and extending the life of the network. It establishes trust between nodes by monitoring packet loss rates and acknowledging messages during the establishment of connections.

The system is based on an IoT kit with a low-cost ESP32 microcontroller, modified by the SSAIL technology and located at a short distance from the data source. Once the kit is installed, it is inaccessible. A working system prototype was used in the laboratory for experimental validation in an IoT scenario. The system was designed to monitor temperature and humidity changes in real time to detect anomalies that could affect the operation of the laboratory. The environment provided challenging operating conditions, and the embedded device pushed the limits of computing power. The IoT kit used an NDS-based unsupervised technique to detect anomalies in system behaviour when operating with an energy budget of less than 1 mW. The system sent identified anomalies as signed transactions to a web server to ensure complete verifiability and non-repudiation. The related work in WSNs emphasises the importance of multidisciplinary approaches that combine hardware innovations, advanced algorithms, and IoT integration. These technologies enable crucial monitoring functions in case of an unforeseen problem with a sensor device or the system being monitored. The optimisation of communication protocols for low power consumption and minimal latency reflects the efficient use of resources, an essential aspect of IIoT systems designed to operate with limited power and bandwidth. 

## 3. Materials and Methods

The need for a real-time IoT monitoring solution drove our research. We developed and tested several practical solutions based on different WSN technologies. These included a protocol stack for SNs such as 6LoWPAN, MQTT-SN, and ContikiMAC; an efficient PSO-DC algorithm for routing; a battery life indicator achieved using the NDS algorithm; an algorithm for calculating throughput when retransmitting encoded packets; a mechanism with which to encode lost packets when throughput meets requirements; an algorithm with which to indicate network hops and beacon intervals; an average radioactivity duty cycle; and a hosting web server with which to store the experimental results of temperature and humidity for environmental monitoring. Our primary research focused on the embedded system of sensor nodes for monitoring temperature and humidity [7]. Some original sensor node antennas and energy harvesters were proposed and prototyped using the SSAIL technology [24,25]. The study in [7] emphasises that SSAIL technology fundamentally changes the design of sensors. It enables the fabrication of antennas and RF-harvesting circuits on various plastics and curved surfaces, which is impossible with conventional methods. 

### 3.1. Integration of Communication Protocols

Initially, we investigated the 6LoWPAN, which is crucial for controlling WSNs. It facilitates data collection and transmission by utilising nearby WSN nodes to forward data to the AP or BS. This protocol, defined by the Internet Engineering Task Force (IETF), uses IPv6 addresses to uniquely address the nodes, which makes it different from other protocols [26,27]. 6LoWPAN is compatible with 802.15.4 devices and other IP networks, such as WiFi, and ensures energy efficiency and reliable data transmission by using a directed acyclic graph (DAG) to route data packets. IEEE 802.15.4 provides link authentication and encryption so that individual nodes can communicate over IP.

Our advanced network architecture, depicted in Figure 1, integrated modified sensor nodes with the SSAIL technology into IPv6 and 6LoWPAN networks through IPv6 tunnelling. This setup offered unlimited addresses and faster and more reliable communication and it also met the sensor node criteria: low cost, short range, small memory, and low bit rate. In this setup, an IPv6 server was connected to the Internet, which tunnelled IPv6 packets into a 6LoWPAN network via an edge node or gateway. The edge node/gateway managed the communication between the 6LoWPAN network and the end nodes, enabling seamless data transmission over low-power, low-bandwidth networks, which are ideal for IoT and IIoT applications. 

This architecture ensured efficiency, scalability, and reliability through the SSAIL technology, which could be adapted to tight spaces and inaccessible locations, as well as the robust network functions of IPv6 and 6LoWPAN. It supported both many-to-one and one-to-many routing modes and allowed idle nodes embedded in the network to remain in sleep mode for an extended time, contributing to overall energy efficiency.

Secondly, we discuss the efficient publish–subscribe framework known as the MQTT-SN protocol, which embodies a typical IIoT architecture tailored for industrial applications. These networks consist of small, battery-powered devices with limited data processing and storage capacities. They operate with limited payload and are often in sleep mode. MQTT-SN protocols offer efficient data transmission and scalability, making them well suited for low-power, low-bandwidth devices, which are often used in IIoT environments. 

The architecture shown in Figure 2 comprises several SN clients that communicate with an SN gateway or SN forwarder via the MQTT-SN protocol. The SN gateway then forwards messages to an MQTT broker via the MQTT protocol, creating a hierarchical communication structure typical of IIoT systems. This setup includes an offline keep-alive procedure for sleeping clients, which ensures that even low-power devices remain connected. 

Integrating various gateways and forwarders demonstrates interoperability within the IIoT framework and enables seamless communication between various devices and systems. The architecture supports scalability, which is crucial for IIoT applications where the number of connected devices can increase significantly. This also means that real-time data processing capabilities are necessary for applications that require an immediate response, such as monitoring and automation. The design emphasises reliability and redundancy. Multiple communication paths and gateways ensure continuous data transmission, even if part of the network fails. In addition, the integration of SN gateways and an MQTT broker complies with the IIoT feature of connecting edge devices to cloud services, enabling advanced analytics and decision-making processes. 

Finally, we considered ContikiMAC, a low-power listening (LPL) protocol used in WSNs and part of the Contiki OS. It is designed to reduce energy consumption in battery-powered sensor nodes by minimising the active mode time of the radio. Its key features include an adaptive duty cycle, low-power listening, packing pacing, and MAC layer optimisation. The sensor nodes alternate between an active state and a sleep state to save energy, with the radio switched down most of the time. The nodes wake up regularly to listen for incoming packets, using the LPL technique to remain in a low-power state while ensuring the necessary communication. This approach strikes an effective balance between energy efficiency and reliable data transmission.

### 3.2. Implementation of PSO-DC Algorithm

By combining 6LoWPAN and MQTT-SN protocols with the ContikiMAC protocol, our method implies that the proposed network system, with its potential to identify relevant events in an industrial IoT environment, promises more efficient energy and coverage planning through improved duty cycle management. Moreover, the PSO-DC algorithm combines particle swarm optimisation with duty cycling mechanisms to optimise energy consumption in WSNs. The method becomes critical when all nodes are in their duty cycle, i.e., when they are sleeping, receiving, and transmitting. In PSO, nodes adjust their positions based on their best-known position and the most prominent position of the neighbouring nodes. This collective movement helps the particles converge to optimal solutions. Minimising latency through congestion-free routing is a significant challenge [26] in WSNs. Duty cycling parameters [27,28,29,30,31], such as duty cycle length, wake-up interval, or the synchronisation scheme, are crucial in WSNs in order to extend network lifetime and minimise energy consumption, especially in applications where nodes are battery-powered, and energy efficiency is critical. The scalability of WSNs in an industrial environment was thoroughly analysed. We evaluated stationary and mobile scenarios in 6LoWPAN networks using new performance evaluation metrics and a novel classification approach.

The proposed PSO-DC algorithm, an innovative solution, is an extension of the classical PSO algorithm specifically designed to dynamically adjust duty cycle parameters in WSNs to improve energy efficiency and network performance. Figure 3 shows an overview of the proposed IIoT architecture based on the PSO-DC algorithm. Our version of the PSO algorithm introduces the following novel improvements:−It dynamically adjusts the duty cycle parameters based on the real-time network conditions and the energy level of the sensor nodes.−It combines traditional PSO with heuristic adjustments to refine duty cycle settings for more precise and contextual optimisation.−It includes an energy-aware fitness feature, prioritising energy conservation while maintaining network performance.

The PSO-DC optimisation algorithm supports WSNs in optimising parameters such as duty cycles, routing paths, or the placement of sensor nodes to boost performance, energy efficiency, and network coverage. Each node must control its duty cycle, including active, proactive, and sleep states. The active nodes are only able to receive data in a WSN. PSO-DC contributes to developing adaptive clustering algorithms or techniques that change the cluster formation in real time according to the input data or changing circumstances in a WSN environment where each sensor node is randomly deployed. The pseudocode for the PSO-DC algorithm dynamically changes the duty cycle parameters through the following process:Initialise the particle swarm with random duty cycle parameters.Evaluate the fitness of each particle using an energy-aware fitness function and network performance metrics. If the particle fitness is better than the personal best, update the personal best. If the particle fitness is better than the global best, update the global best.If the convergence criteria for the individual particles in the swarm are not met, update the velocity and position of each particle using the PSO equations and take into account the adaptive parameter adjustments. In addition, all nodes transmit QoS messages, which are stored in the Q-table and demonstrate the adaptability and versatility of the system. The capability of each node Cn can be evaluated by applying the following expression:(1)$$C_{n}=\omega_{p}P+\omega_{m}M+\omega_{e}E+\omega_{co}C_{o}+\omega_{ac}A_{c};$$
where *P*, *M*, *E*, *C_o_*, and *A_c_* denote the processing power, memory resources, energy constraints, communication overhead, and algorithm complexity of the algorithm, and *ω_p_*, *ω_m_*, *ω_e_*, *ω_co_*, and *ω_ac_* are the weighting coefficients of the above factors, which depend on the specific requirements of each node or the priorities of the network.Check the convergence using predefined criteria (e.g., maximum iterations or minimum improvement). Evaluate the fitness of the updated particle.Adjust the duty cycle parameters of the sensor nodes based on the optimised values obtained from the particle swarm. Return to the optimised duty cycle parameters.

The arbitrary placement of the nodes was initialised by applying the parameters of the node. The CH selection was then set, and the active node of the PSO-DC algorithm was defined in the COOJA simulator as soon as the simulation was started. The system could seamlessly pick up stationary and mobile nodes and send data to the BS. Once the sensors had collected the data, packets were generated and transmitted to the cloud services via a gateway. The active network nodes were selected, while the rest went into a sleep state. Node ID, residual power, and bandwidth were the QoS metrics [12,13] used to determine which nodes were active and which were sleeping. When the minimum energy of the active nodes was almost depleted, the nodes woke up in sleep mode to ensure connectivity. The WSN, which acted as a relay in multi-hop communication, adapted to the needs of the network by efficiently distributing the received packets to nearby nodes. The PSO-DC algorithm used for WSN simulations is described by a flowchart, as shown in Figure 4. It could be adapted to different scenarios.

The above flowchart differs from the traditional PSO algorithm in several ways, reflecting its adaptation to optimising the duty cycles in IIoT applications. This flowchart presents a structured approach to managing sensor nodes that includes different stages and focuses on optimising their performance through energy and bandwidth management and ensuring efficient node operation through Q-table validations. The key differences between the traditional PSO and the proposed IIoT-optimised PSO-DC are listed below:−Initialisation and deployment: The traditional PSO usually starts by initialising a swarm of particles with random positions and velocities in the search space. This collective motion helps the particles to converge on optimal solutions. In contrast, PSO-DC, which is optimised for IIoT, starts with the deployment of random sensor nodes, followed by initialisation and CH selection tailored to the network’s topology and IIoT requirements.−Active mode and node capability: Traditional PSO optimises continuous function by adjusting particle positions and velocities based on fitness evaluations. In contrast, PSO-DC introduces an active mode in which the capabilities of the nodes are evaluated. Nodes with capabilities below a threshold (<0.5) are put into sleep mode to save energy—a step that does not exist in traditional PSO.−Neighbourhood management: In traditional PSO, particles exchange information globally or within a local neighbourhood to converge on the best solution. In contrast, PSO-DC includes specific steps to move neighbouring nodes and evaluate bandwidth and residual energy suitability, adapting the algorithm to network constraints and improving energy efficiency.−Convergence criteria: In traditional PSO, particles converge by minimising a cost function, where the convergence criteria are often set as a predetermined number of iterations or a threshold. In contrast, PSO-DC convergence is based on transmission power, time, and residual energy, which meets the IIoT goals of optimising energy consumption and maintaining network performance.−Bandwidth allocation: Traditional PSO does not explicitly address bandwidth allocation and instead focuses on optimising a mathematically objective function. In contrast, PSO-DC integrates bandwidth allocation as a crucial step that ensures active nodes effectively manage data transmission within network constraints.−Q-table for node justification: Traditional PSO lacks the mechanisms to justify the participation of neighbour nodes in the optimisation process. In contrast, PSO-DC includes a Q-table to justify nodes as neighbours based on specific criteria to improve decision making within the network.−Final steps: Traditional PSO ends with the convergence of particles on the optimal solution. In contrast, PSO-DC ends with identifying active nodes and bandwidth allocation, emphasising the practical application of the optimised parameters within the network.

The PSO-DC algorithm is a sophisticated extension of the traditional PSO algorithm explicitly designed for IIoT environments. It incorporates a comprehensive approach to optimise the duty cycles of sensor nodes by evaluating the node’s capabilities, dynamically adjusting duty cycles, managing neighbouring nodes, and ensuring energy-efficient and reliable data transmission.

### 3.3. Novelty Detection System (NDS)

The NDS is a computational system designed to identify new or unusual patterns that deviate from the norm within a data set. Its value is particularly evident in industry, where the identification of anomalies, outliers, or novel events is critical. The sensor nodes, which are key components, collect data in an industrial plant. They collect average behavioural data from various sources and thus help to detect possible machine faults. On the other hand, the PSO-DC protocol, another outstanding feature, optimises the duty cycle of the sensor nodes in WSNs, focusing on energy efficiency and network longevity. The NDS system’s ability to detect unusual or new patterns in data is a boon for industrial monitoring and network security. With their unique features, both systems improve efficiency and reliability in their respective areas by utilising advanced computing techniques for managing resources and detecting anomalies in order to support industrial operations.

### 3.4. Topology of the WSN and Its Simulation

The WSN was developed using the open-source, highly portable multitasking operating system known as Contiki. We chose the Contiki operating system, programmed in C, because it is suitable for our industrial applications and fulfils the requirements for low power consumption. It can run on microcontrollers with low memory resources, such as 2 kB RAM and 40 kB ROM [2,32]. The integration of Contiki OS into WSN nodes enables the management and configuration of the network and access to the nodes’ data for our industrial applications. One of the most essential advantages of Contiki is the integrated simulation tool COOJA [32]. In our application simulation, the network system evaluated the data parameters of the sensor set selected in Table 1 based on a thorough review to ensure that our results were comparable and based on solid research. In addition to the values from the literature, we conducted initial experiments to validate and fine-tune these parameters. This empirical validation ensured that the parameters accurately reflected realistic conditions and increased the reliability of our simulation results. 

An advantage of the mesh topology is that it facilitates the isolation and detection of network problems and is extensible, while a major disadvantage of some topologies is the hierarchical dependency of the lower parts of the mesh on nodes in higher positions. The proposed model includes three possible network topologies: star, mesh, and tree. The basic idea of the proposed model is to efficiently manage the SN lifetime by controlling energy consumption [33]. Figure 5 shows the simulation model of node placement in the COOJA simulator when using the above parameters and applying our proposed protocols and algorithms, emphasising the protocol’s effectiveness. The simulation settings for static and moving nodes were identical. The main difference was in the mobility model, determined by the PSO-DC algorithm. 

The simulation model visually represents a network of sensor nodes arranged in clusters. The static nodes were modelled in COOJA with 27 clients, 24 of which were regular nodes. Three peer nodes acted as sink nodes and one server (BS) served as a sky mote IPv6. In COOJA, our model was used with the 6LoWPAN, MQTT-SN, and ContikiMAC protocols to represent the mobile nodes, while the PSO-DC algorithm was used to optimise the duty cycle of the mobility model. The server that processes and analyses the data was positioned in the centre of the simulation area. The network was divided into three different clusters. The green cluster represents a star topology consisting of nodes numbered 1 to 6, with node #6 in the centre serving as the sink node; the pink cluster represents a mesh topology consisting of nodes numbered 7 to 12, with node #12 in the centre serving as the sink node; and the blue cluster represents a tree topology consisting of nodes numbered 13 to 27, with node #27 in the centre serving as the sink node. The central nodes in each cluster (6, 12 and 27), the pivotal communication points, are the CHs. They have a great impact by coordinating communication within their cluster and with BS node 28. The network structure, which testifies to an efficient organisation, shows a hierarchical structure. Each cluster worked semi-independently under the coordination of its CH. The clusters were strategically separated from each other, which indicated that the communication between the clusters was handled by the respective CH, ensuring a robust and effective data flow. 

A simulation duration of 2 h was chosen, and the parameters used for the summation are listed in Table 1. After the node data of the simulations were collected, they were processed, various statistical parameters were calculated—e.g., average power consumption and radio duty cycle at different distances—and graphs were generated. This configuration allowed us to analyse the performance of the network and understand how node density and area coverage affect the overall performance of the system. The simulation was run three times to ensure accuracy, and the results were averaged for further consideration. To provide a comprehensive understanding of the data trends and an effective comparison of the various scenarios, the results were plotted using Matplotlib. Furthermore, the proposed NDS identifies all anomalies. The results show that the proposed framework is one of the suitable solutions for industrial IoT, providing low latency and an efficient communication performance. 

### 3.5. WSN Deployment and Its Visualisation

Simulations are a valuable tool in WSN projects as they enable the exploration of the design space and the identification of potential problems. In systems engineering, it is common to use more extensive simulations and smaller initial implementations as a technique when errors can have significant costs and consequences. This approach provides a harmonious blend of theoretical modelling and practical, actionable data from real-world operations. The experimental setup for the comparison was consistent and fair, as the same simulation environment and parameters were used for all algorithms.

The initial experiments, which formed the essential basis for our research, were conducted under the assumption of static conveyor belts. Figure 6 provides a clear schematic representation of the initial planning and visualisation of this concept, which was probably intended to be part of a simulation of WSN deployment in real time to see how the sensor nodes were organised on a separate conveyor belt [34] and managed within a WSN that eventually grew. The nodes were arranged in a grid, which facilitated the visualisation of the network’s topology and their relative positions. A long rectangle represents the conveyor belt. The nodes along the conveyor belt indicate the positions of the stationary nodes, and the objects moving along the belt are moving nodes. The sink node near the BS and the conveyor belt was static, while the other nodes were attached to the belt that moved with the objects. The nodes near the periphery of the belt, forming a cluster, have different roles when considered as objects on the belts, e.g., as intermediaries for data transmission to the CH. The decision to simulate or deploy sensor nodes depended on the project’s phase, resources, and specific objectives.

Interference in the IoT network affects the accuracy of event detection and poses a challenge due to interference from neighbouring nodes, leading to collisions and increased packet loss. A well-designed WSN project often involves thorough simulation and step-by-step real-time deployment to maximise effectiveness and efficiency. In the laboratory, an ESP32 that had been specially modified with the SSAIL technology was used to test protocols and communication platforms. The market currently offers a wide range of new IoT-related components that offer enormous opportunities for creativity. Figure 7a shows the schematic representation of the network’s topology, demonstrating the position of the BS, the receiver nodes, and the SN in relation to each other and their distances from the BS, and Figure 7b shows the topology of the network in the simulator used in the laboratory for initial trials before the sensor nodes are deployed in the FTMC laboratory to lower the cost of deployment and test the performance of the networks in practice. We characterised various aspects of energy demand in static and mobile-assembly-line scenarios and analysed the average power consumption, data transmission latency, radio duty cycle, and the average number of hops of the networks after optimisation.

The hierarchical connections indicate the structured use of the nodes to enable efficient data aggregation and transmission. The nodes are grouped under receivers, indicating clustered communication. Receiver 1 and Receiver 2 correspond to the CH. The nodes under “Receiver 1” (nodes 1, 2, and 3) are analogous to the nodes in a cluster in the simulation in Figure 7b). Similarly, the nodes under “Receiver 2” (5, 6, 7, 10) form another cluster. Nodes 4 and 8 are intermediate nodes that connect different network parts to the BS and improve communication efficiency and coverage. Nodes 9 and 10 also act as intermediate nodes under Receiver 2. The results of the comparative analysis are presented in Section 4. In Section 5, we discuss in detail how our proposed algorithm outperforms the existing algorithms.

### 3.6. Mobility Model

The impact of mobility on the performance of WSNs, especially in mobile applications such as production unit monitoring, can be significant and challenging. When nodes relocate, this leads to changes in the network structure and requires a recalculation of routing paths. This can lead to more data traffic in the network, a drop in the data transfer rate, unsuccessful packet delivery and, ultimately, increased energy consumption. All of the above problems are the subject of our research. A significant advance in our research was the focus on the impact of moving conveyor belts on the system performance, especially on node density and area coverage. These factors were taken into account, and simulations were performed in three scenarios with identical grid-based areas to the initial set, with dimensions of 25 m × 25 m, 50 m × 50 m, 75 m × 75 m, and 100 m × 100 m. The PSO algorithm is popular and widely used by other researchers [19,20,21,22,27,30]. However, our simulations were performed based on our proposed PSO-DC optimisation algorithm, and the nodes were designed to move according to the PSO-DC optimiser so that we could observe and analyse the performance of the system under these dynamic conditions and the packet delivery ratio (PDR).

The PSO-DC mobility model randomly placed the nodes in predetermined simulation areas at the beginning of the simulation. Initially, each mobile node remained stationary for a randomly calculated stop time, with values evenly distributed between 0 and 60 s. After the specified time, the node moved to a random location within the simulation area, with its speed evenly distributed between a minimum and a maximum value. The node then moved towards the desired destination at its assigned speed. Upon reaching the destination, the node stopped temporarily for the specified duration before repeating the process again. It selected a different combination of speed and destination and continued the movement after the pause. In the simulation, the mobile nodes were uniformly assigned a speed of 1 to 2 m/s. In addition, the mobile nodes may have different signal strengths and interference levels, further complicating the maintenance of a consistent connection. Therefore, it was vital to consider the impact of mobility on the overall performance of WSNs designed for mobile applications. In addition, the NDS detected any anomalies in the network functions.

## 4. Results Analysis of the Simulation and Experimental Investigations

This section contains tables, graphs, and statistical analyses to illustrate the differences in WSN performance. The results show that the proposed framework is suitable for stationary and moving situations and provides an effective communication performance and low latency. The results show the advantages of using the PSO-DC instrument for mobile and static SN routing. It can handle dynamic topologies and improve the overall performance of the network. In addition, NDS identifies new or unusual patterns in data that deviate from established norms. The results show that the proposed framework for IWSNs helps to detect specific events, such as temperature anomalies, speed variations, overload conditions, maintenance alerts, and conveyor belt tensions in production lines to ensure smooth operation, quality control, and safety. 

### 4.1. WSN Simulation and Analysis

The aim of the simulation was to validate the proposed method in industrial applications. The simulation allowed us to accurately compare the performance metrics of the network of stationary and moving nodes with the parameters listed in Table 1. The simulation made it possible to identify the most sensitive parameters affecting the overall performance. Using the COOJA simulator allowed us to evaluate the system in a regulated environment, reducing the cost and complexity associated with installing the system in real applications. In addition, the simulation results could be easily duplicated and validated, which significantly increased the accuracy of the results. Simulation was crucial to our research methodology because it allowed us to control data acquisition from the network via the BS. The serial connection functions allowed us to connect or disconnect lost nodes, ensuring real-time network communication. We used various simulation parameters such as reporting intervals, randomness, hop-by-hop retransmissions (0–31 hops), and reports per hour to set up data collection. The Contiki platform sent commands to the nodes, such as stop and reconnect, to facilitate the control of the WSN during data collection. 

We simulated three network topologies: star, mesh, and tree. Using the parameters in Table 1, we determined the average number of hops the nodes required to communicate information to the BS in order to estimate the number of nodes in real environments. We applied the PSO-DC algorithm, a reinforcement learning (RL)-based network application [36,37], to allow the SNs and the BS to track and respond to their environment. This significantly improved the latency and throughput of the network, especially in larger networks. In the first scenario, a single stationary sink node was used to collect sensor data from 27 stationary nodes. The nodes were distributed over 4 square areas with the following dimensions: 25 × 25 m^2^, 50 × 50 m^2^, 75 × 75 m^2^, and 100 × 100 m^2^. The same four square areas were simulated in the second scenario with the application for mobile nodes. In this scenario, a single stationary BS node was used to collect sensor data from 27 mobile nodes. In this case, the data were collected after swarm optimisation where the optimal criterion was the duty cycle, and the best position of the neighbouring nodes was found. Figure 8 compares the first set of results of the average end-to-end PDR between the mobile nodes and the BS of the two scenarios and dissemination areas. The latency was calculated as an average value over three runs, each lasting 2 h. 

In addition, the results considered the end-to-end delay, i.e., the average time it took for a data packet to travel over the network from source to destination. Node latency was directly related to the number of hops: it indicated the number of hops between a node and the destination. The node latency directly reflected the PDR: the lower the latency, the higher the PDR, and the higher the data latency, the lower the PDR. If all nodes were fixed, as in a stationary conveyor belt, all packets sent from one node to the BS followed the same path, so that the number of hops was fixed for all delivered packets. In a mobile conveyor belt scenario, the network topology constantly changed so that packets sent from a node to the BS followed different paths and had a different number of hops. This dynamic environment led to a complex network architecture with fluctuating hop counts for different packets. 

Our investigations address critical aspects of network connectivity, i.e., the fraction of all SNs that can transmit at least one packet to the BS, with values ranging from 100% to 0%. This key metric consistently proved to be more meaningful in simulation areas in the fixed scenario than the PDR in the mobility scenario. This result emphasised the importance of network density and its impact on the extent of congestion. For example, in a densely populated network with a grid size 25 × 25, switching to a less dense network led to an increase in PDR and, therefore, less congestion. However, for a 100 × 100 network, the PDR was lower, indicating a lack of path information from all nodes to the BS, which could cause some nodes to become isolated due to the low connectivity of the graph. Moreover, the poor PDR for mobile nodes in the 50 × 50 and 75 × 75 scenarios gave a better result than was obtained for static nodes [2] since the network topology changed frequently due to the mobility of these nodes. The results we obtained from the simulation with the PSO-DC optimisation algorithm are shown in Figure 9. They illustrate this complexity and show that IMR, with its extended network, has a very high average hop count [37]. The averages of the three optimisation strategies—RL, iterative minimum residual (IMR), and optimal network—were presented and analysed. Even RL, which aims to minimise the number of hops, has a higher average hop count than the optimal mesh. It is intriguing that RL can achieve a low number of hops without the switch capabilities of conventional routers, which increase the complexity of the network architecture. 

To evaluate the effectiveness of each node, assessed according to the formula (1), and the entire WSN, we considered the number of packets received per node, the average radio duty cycle, and the average power consumption. Figure 10 shows the number of total and duplicate packets per node, where the x-axis represents the node numbers (1–26) and the y-axis represents the number of packets (0–30). Blue bars show the total number of packets, while orange bars show the duplicates. Central nodes received more packets due to their role in data aggregation, while peripheral nodes received fewer packets. Nodes in areas of high interference may have more duplicates as they are retransmitted more frequently. High numbers of duplicates occurred at nodes that were part of multiple paths or in areas of high interference, indicating congestion, collisions, or inefficient routing.

An important result is that node #10 received the largest number of packets (about 26) without duplicates, indicating its central role in the network as a key relay or hub. Node #5 and node #21 also had high packet reception, with node #5 receiving about 15 packets and node #21 receiving about 18 packets, indicating their important role in data aggregation and forwarding. Node #8 had the highest number of duplicates (about 6), probably due to collisions or redundant paths, indicating its position on a critical path with multiple routes. Nodes #1, #2, #4, #7, #8, #15, #19, #20, and #21 also had notable numbers of duplicates, indicating their central role and higher traffic volume or interference. Nodes #25 and #26 had the lowest packet reception, with node #26 receiving only one packet and node #25 receiving about two packets, indicating their peripheral position in the network or weaker connections. The diagram shows the distribution of packets and duplicate packets across the nodes, reflecting their role and position and the routing efficiency of the network. A high packet reception indicates important nodes, while many duplicates indicate overlapping routes or congestion. Peripheral nodes have lower reception due to their limited role. By understanding these dynamics, it is possible to optimise the network design and improve the efficiency of data transmission. Overall, Figure 9 illustrates the hops required for data packets under different optimisation techniques that are directly related to network latency. Table 2 shows the effects of the different optimisation methods on the number of hops, the reasons for the hops, and their effects on latency.

The table shows that the network without optimisation had the highest latency when the number of hops increased. IMR optimisation improved reliability but still had a relatively high latency. RL optimisation provided a balance, with fewer hops and lower latency. The optimal optimisation offered the best performance, with the fewest hops and the lowest latency. It reduced the average latency from 150 ms to 50 ms and lowered the packet loss rate from 5.0% to 1.0%.

Figure 11 shows the percentage of time that nodes 1–26 spent listening and transmitting radio waves. The duty cycle, which ranged from 0% to 12%, was divided into blue (listening) and orange (transmitting) bars. Node #12 had the highest duty cycle at 10%, which indicates its central role. Nodes #10 and #11 also had a high-duty cycle and balanced each other out when listening and transmitting. Most nodes listened more than they transmitted. Nodes #2, #6, #7, #14, #15, #20, and #24 had a low-duty cycle, with nodes #2 and #20 showing minimal activity. Node #26 shows a peak in sending activity despite its low packet reception, which indicates high activity. This visualisation illustrates the different levels of activity and communication patterns across the network.

Nodes with high-duty cycles, such as nodes #10, #11, and #12, were at the centre of communication in the network and acted as intermediaries or relay points. Nodes on the periphery, such as node #2 and node #20, had lower duty cycles, indicating less involvement. The balanced distribution of listening and sending tasks among the central nodes indicates active data exchange. Peaks in the duty cycles of nodes, such as node #26, may be due to certain events such as rerouting or congestion processing. Nodes with high utilisation consume more energy and require efficient energy management strategies such as duty cycles or load balancing to ensure network longevity and reliable communication. Understanding these patterns is essential to optimising network performance and managing energy consumption. 

Figure 12 shows the average power consumption of nodes 1–26 in low-power mode (LPM), CPUs, radio reception, and radio transmission. The power consumption ranged from 0 to 1.8 mW. Each node had four stacked bars representing these modes. Most nodes consumed 0.8–1.4 mW. Node #20 had the highest consumption of 1.4 mW; this was mainly due to radio transmission. Nodes #12, #18, #21, and #22 also had high power consumption due to radio transmissions and CPU activities. Nodes #2, #3, #4, #5, and #7 had balanced power consumption in all categories. Radio transmission was important for nodes #12, #20 and #21. This profile helps to optimise the energy consumption of the network.

### 4.2. Deployment of WSN under the Conditions of Technological Laboratory

The initial trials aimed to reduce deployment costs and improve cost efficiency in real-time conveyor systems. To this end, ten modified ESP32 controllers using the SSAIL technology were placed in the FTMC lab, symbolising the stationary nodes in conveyor belts. Nodes that were located close to the BS had lower energy requirements, while nodes further away required more power, which affected their energy consumption. USB-powered nodes ensured continuous operation and reliable data transmission at critical points, while the battery-powered nodes were optimised for energy efficiency and mobility. Efficient power management was essential for battery-powered nodes, especially in outdoor areas or remote scenarios without wired power sources. 

Understanding the duty cycle of a battery-powered ESP32 is crucial to managing its battery life. The main factors affecting battery life were capacity, power consumption, and duty cycle. A typical 18,650 lithium-ion battery has a capacity of 3000 mAh (3.7 V). The ESP32 consumed 160–240 mA when processing and transmitting data via WiFi. Table 3 summarises the battery life calculations, assuming that sensor measurements were taken every 10 min. 

The ESP32 with a 3000 mAh battery, which was woken up for 5 s every 10 min, lasted about 67.5 days. Adjusting the duty cycle, battery capacity, or power consumption had a significant impact on battery life. At critical points, USB-powered nodes increased network reliability and data throughput by providing a stable and continuous power supply, enabling frequent data transfers and constant operation. The strategic use of battery-powered nodes required efficient power management, especially for those nodes far from the BS, which required higher transmission power. Effective power management ensured long-term functionality and reliability. The balance between USB and battery power sources provided the network the flexibility and stability essential for continuous and efficient data communication. Temperature and humidity were regularly monitored with DHT11 sensors from March 2023 to February 2024. The special approach [7] was used for data collection, where high-level PHP applications were hosted using the MySQL 8.0.36 database and the data were visualised in real-time graphs on the web server. The average, minimum and maximum temperatures are plotted in Figure 13. Similarly, the humidity data are also recorded in Figure 14.

Figure 13 shows the monthly temperature data collected from March 2023 to February 2024 using the WSN installed in the FTMC laboratory. The temperature was highest in August at 38 °C, while it was the lowest in January at 11 °C. The temperature fluctuated greatly in summer (June–August) and was higher in the warmer months. In winter (December–February), the maximum temperature was lower and the average and minimum values were more balanced. The graph clearly shows the monthly temperature trends throughout the year and proves that WSN can seamlessly provide an uninterrupted data supply of the climate conditions in a technological room over a long period of time.

Figure 14 shows the monthly humidity data collected from March 2023 to February 2024 using the WSN installed in the FTMC laboratory. In August, the humidity was highest, at around 70%, while it was lowest in July at 48%. Humidity fluctuated greatly in summer (June–August) and was higher in the warmer months. In winter (December–February), the maximum humidity was lower and the average and minimum values were more balanced. The diagram clearly shows the monthly trends in humidity throughout the year. 

Our efforts to optimise the WSN significantly improved the performance of the network. Energy consumption was reduced, which increased efficiency and extended the battery life of the mobile nodes. Fewer duplicate packets improved data throughput and minimised unnecessary transmissions, which increased the overall efficiency of the network. Table 4 shows these significant improvements. 

Optimised energy management in mobile nodes increased efficiency and reliability. Nodes near high traffic areas operated more efficiently, with fewer data collisions and lower energy consumption. These improvements contributed to a more robust, reliable, and efficient WSN.

## 5. Discussion

The integration of 6LoWPAN [1,8,9,10,39] and MQTT-SN [21] in low-power wireless networks improves the reliability and efficiency of IoT communication [5,28,29,40], especially in resource-constrained scenarios. This protocol ensures effective event-driven communication [26,27], which is crucial for IoT devices in industrial environments. The implementation of SSAIL technology further enhances this integration by enabling the development of miniaturised, energy-efficient sensors with integrated harvesting circuits and antennas [16]. Sensor nodes that are modified with SSAIL in combination with ContikiMAC’s [23] energy-efficient [13,14,15,22,27] connectivity and routing protocol [11,12,13,14,17,33,41] provide promising results to ensure longer lifetime and good efficiency, which are among the desired solutions in the field of IIoT applications [3,4,30,42,43]. The convergence of these protocols and technologies, including SSAIL, provides a secure, reliable, and efficient solution for IoT communication in industrial practice and other low-power applications [44], opening the way for future advancements [5,6,35]. They can be further improved by integrating various advanced technologies and approaches to make them more efficient and scalable.

The 6LoWPAN protocol is essential for efficient data collection and transmission in WSNs and utilises IPv6 addresses for improved scalability and reliability. This protocol, in combination with the modified SSAIL sensor node, enables cost-effective, low-power, and reliable communication suitable for constrained environments. The MQTT-SN protocol further enhances IIoT systems by ensuring efficient publish–subscribe communication, supporting scalability and integrating edge devices with cloud services for advanced analytics. Finally, the ContikiMAC protocol significantly reduces energy consumption through its low-power listening approach, balancing energy efficiency with reliable data transmission. Together, these protocols provide a robust framework for optimising WSNs and offer a comprehensive solution for modern IIoT systems that focuses on efficiency, scalability, reliability, and energy savings, making it superior to other available WSN systems.

The performance of our proposed framework was rigorously evaluated through experiments in static scenarios where the nodes were fixed at one point. The results for the mobile scenario were derived from a simulation that showed how mobility affects performance. After starting the simulation, the mobile nodes were randomly distributed in predetermined areas based on the PSO-DC algorithm [11,12,13,14,15,31]. The results showed that node density [45,46] and area coverage significantly influenced the system’s performance in the static state. The results of averaging three 2-h runs determined the end-to-end PDR [47,48,49] of the nodes to the BS. The results showed that the PDR increased in parallel with the node density, indicating that a denser network performed better. Similarly, the PDR increased in parallel with the coverage area, indicating that a larger coverage area led to better network performance. 

The simulations were performed with a mobile node speed of 1–2 m/s. The hop count metric evaluated the reliability of the network in transmitting data packets from their origin to their destination. The results showed that the number of hops increased proportionally with the mobility of the nodes [50] and a decrease in PDR. The challenge of dealing with rapid changes in the network architecture, which could lead to a loss of connectivity and non-delivery of data [45] when mobile sensor nodes move beyond the transmission range of their parent nodes, was a key area where our research could provide valuable insights. The PSO-DC algorithm has a potential memory for the neighbouring nodes, which helps to create robust network architectures. The simulation results of the experimental study of the SSAIL-modified WSN in the FTMC technology laboratory show that the SN worked for about 68 days without interruption.

The simulation results validated our proposed WSN topology and showed its potential for efficient communication performance and minimised latency in both static and mobile scenarios. The practical importance of an optimal solution for static and mobile scenarios in industrial applications was supported by the findings on the impact of node mobility on network performance, which necessitated the development of strategies to maintain connectivity and data integrity. The simulation results of the COOJA simulator [32] could be applied to various industries where conveyor belt applications [34] provide a comparison of the performance of mobile and static nodes in realistic environments. The above-mentioned capabilities of the proposed technique could significantly reduce the complexity and cost of the deployed WSN in further implementation.

The specification of SN applications [51] helped the QoS [12,13] to overcome unique challenges such as resource constraints, unbalanced traffic and its varieties, data redundancy, energy optimisation [15,17,18], scalability, multiple sinks [52], and packet criticality. Our proposed method outperformed other clustering protocols [19,20,53] in terms of network longevity, throughput, and adaptability, as measured by various industry-standard metrics. This mode ensured that the network operated efficiently, minimising energy consumption and maximising performance.

Our future research will dive into the unexplored territory of IoT networking and explore the cutting-edge technologies of 5G connectivity [54,55], blockchain technology [56,57] for IoT security [36,37], and the integration of AI into network management. This unique approach not only explores the potential impact of these trends on the evolution of IoT networking, but also evaluates the proposed architecture in terms of incorporating IoT devices into industrial applications using two real-world cases—stationary conveyors and movable assembly lines—which provide insights into practical applications. Inefficiencies or design decisions in the routing protocol can lead to redundant paths, especially in dense networks or those with high mobility, which we will emphasise in our future work. However, the seamless deployment of WSNs in real-world environments, taking into account environmental variability, different security requirements, and network scalability, still requires further research. Future work will also include further improving power management strategies, exploring advanced routing protocols, and applying our optimisation techniques in various real-world scenarios to validate and extend our results.

## 6. Conclusions

A new approach to the integration of WSN with 6LoWPAN, MQTT-SN and ContikiMAC protocols in an industrial IoT system was proposed and validated. The techniques proposed in the recent work, using the COOJA simulator and PSO-DC optimiser, enabled the introduction of the metric of quantitative WSN evaluation of duty cycle parameters in the early stage of industrial WSN development, which was used in practical projects. This can have a positive economic impact on the development of industrial WSN applications. The main valuable findings of the recent work are summarised in the following statements:−We undertook the pioneering integration of 6LoWPAN and MQTT-SN in an industrial IoT system.−A new NDS method, based on event detection and other realistic factors, was proposed.−The estimation of connection quality at the receiver node was analysed by counting the number of consecutive packet losses, duplicates, and surpluses in each connection.−A metric was proposed for quantitative WSN evaluation using the quality parameter weighting method.−A schematic representation of initial WSN deployment planning and visualisation in conveyor systems was proposed.−We analysed latency in data transmission and assessed the average number of hops of WSN optimisation based on four quadratic domains, considering static and mobile conveyor belt scenarios. The results of the analysis of the average duty cycle and various optimisation techniques were presented.−A simulation model of the network topology and its analysis of the mobility model were presented.−We presented the application of the simulation results of the WSN Contiki COOJA to evaluate the battery life, security, reliability, and possible traffic bottlenecks of the network.−The estimation of the average power consumption of SNs in different operating modes and a comparison of the results of the optimisation of WSN components were presented.−An unsupervised NDS model was implemented and investigated using a constrained device at the edge of the network, and the results of the experimental investigation of the SSAIL-modified WSN in the FTMC technology lab were presented.

This research lays the foundation for further investigation of the performance and scalability of the industrial IoT framework and promises real-world solutions that can significantly improve industrial operations.


## Figures and Tables

**Figure 1 sensors-24-04881-f001:**
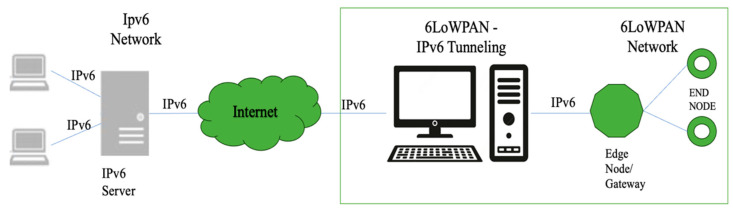
LoWPAN system architecture.

**Figure 2 sensors-24-04881-f002:**
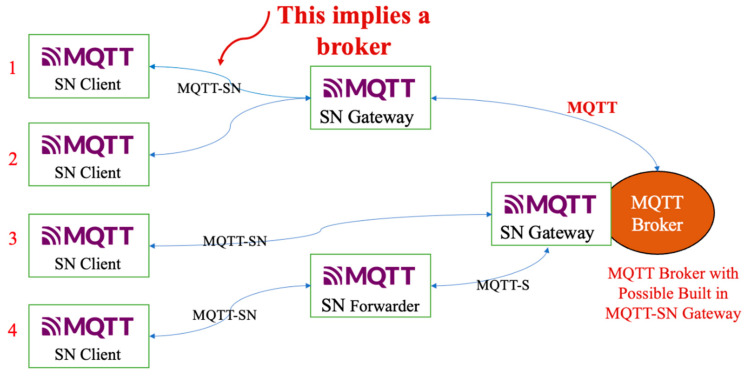
MQTT-SN system architecture.

**Figure 3 sensors-24-04881-f003:**
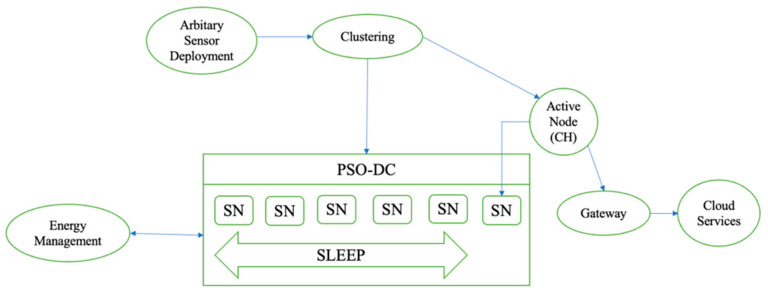
Proposed IIoT architecture based on PSO-DC algorithm.

**Figure 4 sensors-24-04881-f004:**
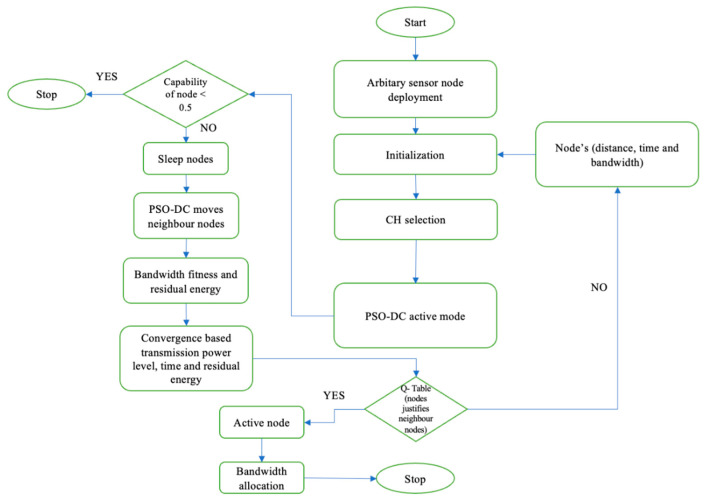
Flowchart of the proposed PSO-DC algorithm.

**Figure 5 sensors-24-04881-f005:**
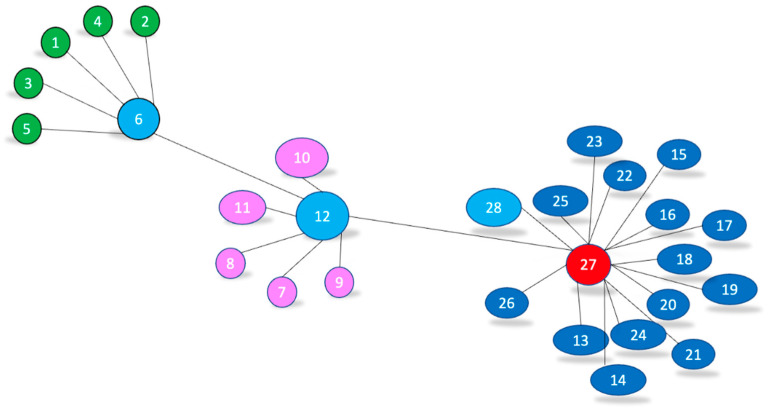
Simulation models of network topologies: star, mesh, and tree.

**Figure 6 sensors-24-04881-f006:**
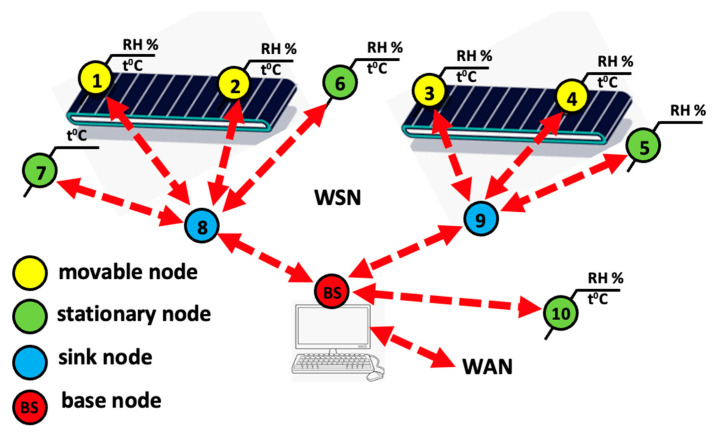
Schematic diagram for initial planning and visualisation.

**Figure 7 sensors-24-04881-f007:**
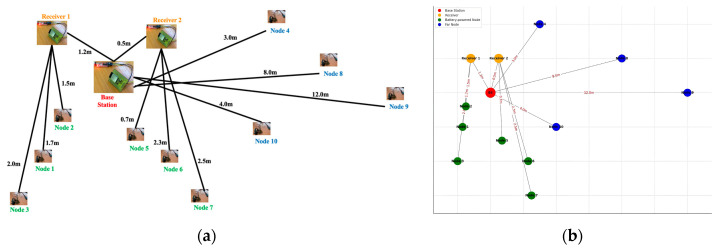
Topology of the WSN (**a**) deployed in the FTMC laboratory (**b**).

**Figure 8 sensors-24-04881-f008:**
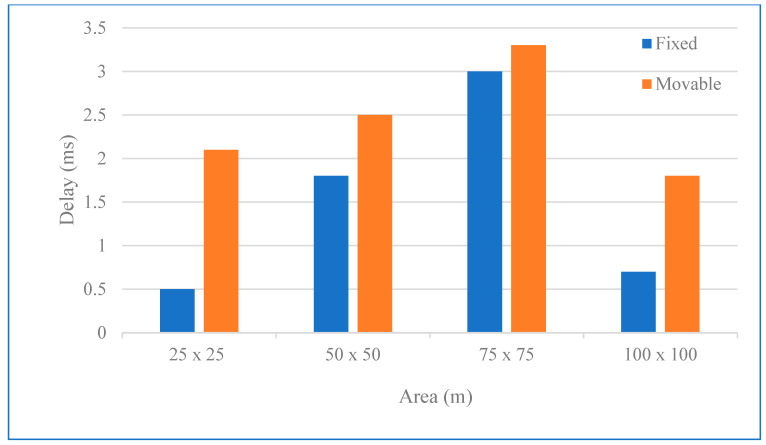
Data delivery latency for fixed and movable nodes on four square areas of 25 × 25 m^2^, 50 × 50 m^2^, 75 × 75 m^2^, and 100 × 100 m^2^.

**Figure 9 sensors-24-04881-f009:**
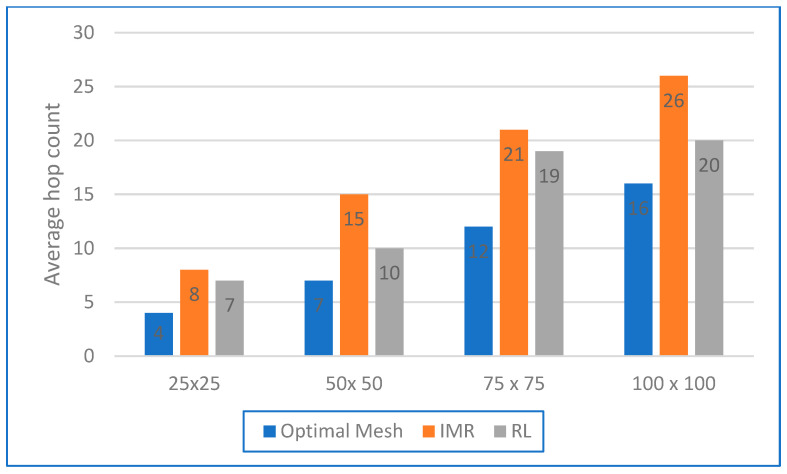
Average hops count in the networks after WSN optimisation with different strategies: optimal mesh, IMR, and RL network in the areas of 25 × 25 m^2^, 50 × 50 m^2^, 75 × 75 m^2^, and 100 × 100 m^2^.

**Figure 10 sensors-24-04881-f010:**
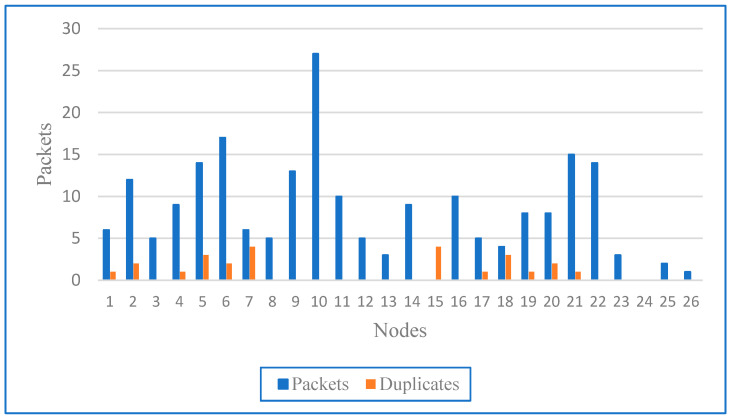
Number of received packets per node versus nodes.

**Figure 11 sensors-24-04881-f011:**
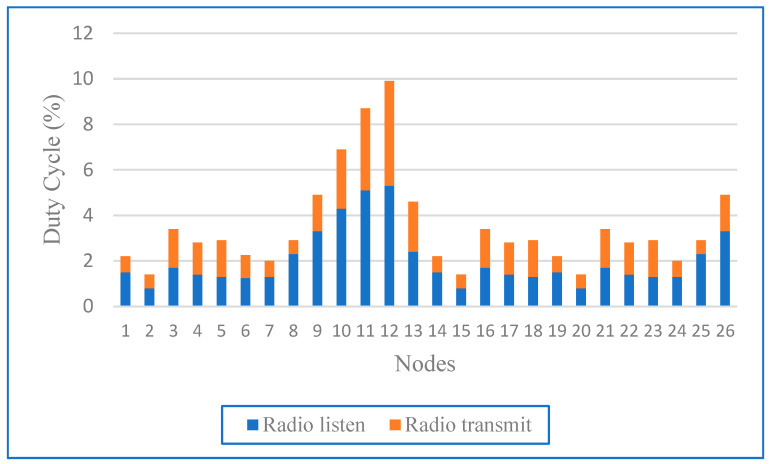
Average radio duty cycle.

**Figure 12 sensors-24-04881-f012:**
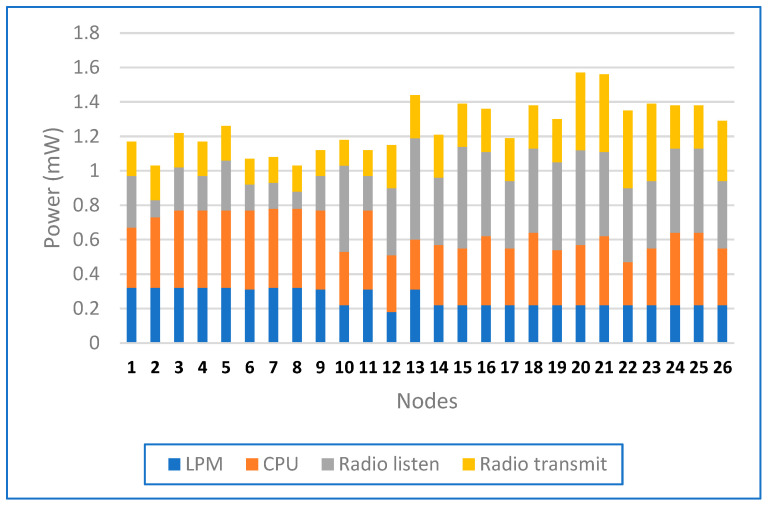
Average power consumption of the nodes in different operation modes and components: LPM, the consumption of the CPU, radio listen, and transmission modes.

**Figure 13 sensors-24-04881-f013:**
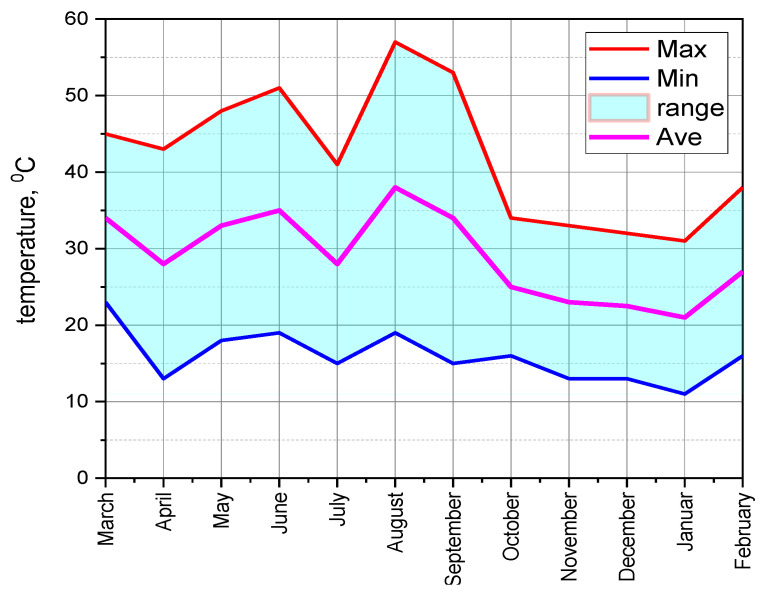
Temperature readings collected by the WSN in the FTMC laboratory.

**Figure 14 sensors-24-04881-f014:**
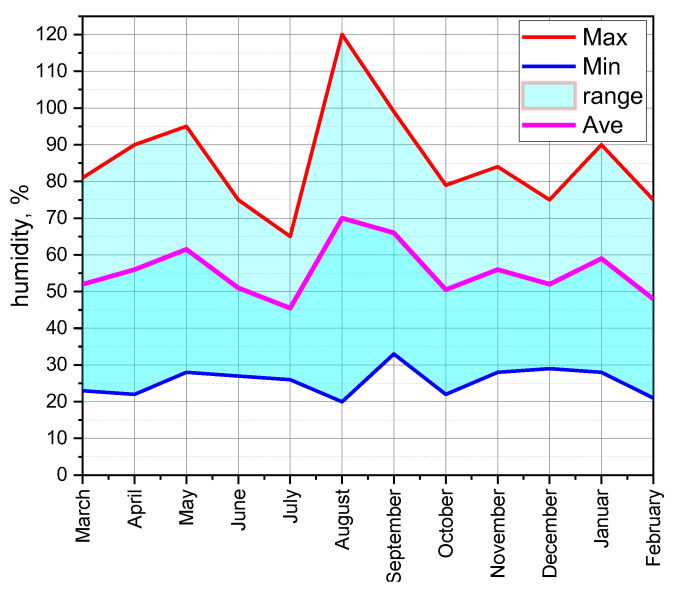
Humidity readings collected by the WSN in the FTMC laboratory.

**Table 1 sensors-24-04881-t001:** Parameters for the simulation in the COOJA simulator.

#	Parameter	Value	Source/Justification
1	Number of motes	24 + 3 sinks	Common practice in WSN studies, ensures adequate data collection points [4,5,15,29]
2	Type of routing	IPv6	Ensures compatibility with modern network protocols [4,5,11,12,13,14,23,30,33]
3	Standard used	6LoWPAN IEEE 802.15.4	Suitable for low-power and lossy networks [6,8,9,10,17,18,30,31]
4	Mote type	Sky mote	Reliable performance; compatible with Contiki OS [19,32,34,35,36]
5	Mote distribution	Grid	Ensures even coverage and connectivity [19,20,21,22,27,30]
6	Tx and Rx success ratios	1.0	Defines a primary performance metric [14,16,17,18,27,30,31,34,36,37,38]
7	Reporting intervals	60 s	Ensures a balance between timely data collection and energy consumption [26,27,28]
8	Hop-by-hop retransmissions	0–31	Enables robust data transmission without excessive overhead [15,28,29,30,32,33,34]
9	Duration	2 h	Provides sufficient time to observe network behaviour [32]

**Table 2 sensors-24-04881-t002:** Comparison of the different optimisers.

Optimisation Method	Number of Hops	Reason for Hops	Effects on Latency
Without optimiser	10	150	5.0
IMR optimisation	7	120	3.5
RL optimisation	4	80	2.0
Optimal optimisation	2	50	1.0

**Table 3 sensors-24-04881-t003:** Energy consumption and lifetime of the Li-18650, 3000 mAh battery.

	Mode	Current, mA	Mode Duration per 1 Cycle, s	Charge per 1 Cycle, mA∙s	Cycle Duration, s	Average, mA
Node ESP32+DHT11 Sensor	Active	200	5	1000	600	1.85
	Modem sleep	10	5	50		
	Light sleep	0.8	0	0		
	Deep sleep	0.1	595	59.5		
Total battery life: 67.5 days						

**Table 4 sensors-24-04881-t004:** Optimisation of WSN components.

Aspect	Before Optimisation	After Optimisation	Improvement
Energy efficiency	200 mA (active mode with WiFi)	100 mA (optimised mode with WiFi)	Reduction in energy consumption by 50%
Battery life	15 h	67.5 h/days	350% increase in battery life
Reduction in duplicated packets	50 duplicated packets/hour	10 duplicated packets/hour	80% fewer duplicated packets
Mobile nodes	10 h (3000 mAh battery)	30 h (3000 mAh battery)	200% improvement in operating time
Stationary nodes	5 W average power consumption	2.5 W average power consumption	50% increase in efficiency and reliability
Nodes close to the conveyor belt	100 data collisions/hour	20 data collisions/hour	80% fewer data collisions, improved

## Data Availability

The original contributions presented in the study are included in the article, further inquiries can be directed to the corresponding author.

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
