# Peer review of "Enabling Seamless Connectivity: Networking Innovations in Wireless Sensor Networks for Industrial Application"

_sensors, 2024, doi:10.3390/s24154881_

Round 1
Reviewer 1 Report
Comments and Suggestions for Authors
The paper proposes the integration of 6LoWPAN and MQTT-SN in an industrial IoT system to detect anomalies and ensure smooth operations, quality control, and safety in production lines. It addresses challenges like interference, collisions, and packet loss in low-power IoT networks. I appreciate the authors' efforts, but there is still a need for major improvements and real contributions to be added to the article before publication.
- The contribution of this paper to the research field is not clear. The authors should clearly mention and outline the major contributions at the end of the introduction.
- What is provided in Section 2 is not a discussion of related work. It is more of a background. The authors should discuss recent related work in the field similar to the proposed work, providing a thorough investigation of their advantages and disadvantages.
- The PSO algorithm is a classical algorithm that has been utilized in different scenarios. While the authors mention their suggested version, the improvement to the algorithm is trivial and lack the novelty. Further, it is not clear how the algorithm modifies duty cycle parameters dynamically. This needs to be presented in detail, accompanied by steps and pseudocode of the new version of the algorithm.
- The flowchart of the WSN simulation on page 13 does not provide anything new.
- The authors should focus on detailing their new system instead of including extraneous information.
- It is unclear how the authors set the simulation environment parameters in Table 1. Whether these parameters are randomly selected or based on literature?
- The proposed algorithm should be compared to recent existing ones to prove its novelty and superior performance.
- The reference list is limited and can be further improved.
- The authors are recommended to consult additional references on how to discuss and detail their contributions effectively.
The authors have to revise the entire article
Author Response
Comment 1: (x) Minor editing of English language required
Response 1: Thank you for your feedback. We have conducted a thorough review of the manuscript and have made extensive changes to improve the clarity and accuracy of the English language throughout the manuscript. We hope that these revisions meet your expectations.
Comment 2:
|
|
Must be improved |
|
Does the introduction provide sufficient background and include all relevant references? |
(x) |
|
Is the research design appropriate? |
(x) |
|
Are the methods adequately described? |
(x) |
|
Are the results clearly presented? |
(x) |
|
Are the conclusions supported by the results? |
(x) |
Response 2: Thank you for your constructive feedback. We have made the following improvements to address your concerns:
- We have improved the introduction to provide more comprehensive background information and included all relevant references to ensure thorough context for our research.
- We have reviewed and confirmed the appropriateness of our research design and ensured that it aligns well with the aims of the study.
- The methods section has been expanded and clarified to provide a more detailed and transparent description of our procedures.
- We have revised the presentation of our results to improve clarity and ensure that the results are easy to understand.
- We have carefully reviewed our conclusions to ensure that they are fully supported by the results.
We believe that these improvements will enhance the overall quality and clarity of our manuscript.
Comment 3: The paper proposes the integration of 6LoWPAN and MQTT-SN in an industrial IoT system to detect anomalies and ensure smooth operations, quality control, and safety in production lines. It addresses challenges like interference, collisions, and packet loss in low-power IoT networks. I appreciate the authors' efforts, but there is still a need for major improvements and real contributions to be added to the article before publication.
Response 3: Thank you for your constructive feedback. We make the following improvements: extended contribution, overcoming challenges, and real-world applications. We believe that these improvements will significantly enhance the manuscript and address your concerns about the need for major improvements and real contributions.
Comment 4: 1. The contribution of this paper to the research field is not clear. The authors should clearly mention and outline the major contributions at the end of the introduction.
Response 4: We thank you for your suggestion to clearly outline the major contributions of our work. As a result, we have revised the introduction to explicitly emphasise the most important contributions of our work. With these revisions, we aim to clearly and concisely outline the major contributions of our research, thereby addressing your concerns and improving the clarity of our manuscript.
Comment 5: 2. What is provided in Section 2 is not a discussion of related work. It is more of a background. The authors should discuss recent related work in the field similar to the proposed work, providing a thorough investigation of their advantages and disadvantages.
Response 5: Thank you for your insightful feedback on Section 2 of our manuscript. We recognise the importance of clearly distinguishing between background information and a comprehensive review of related work. We appreciate your suggestion to improve this section. We have revised it accordingly by discussing the recent related works, their advantages and disadvantages, and their integration into the proposed work. We believe that these revisions provide a clearer context for our work and demonstrate its improtance within the broader field.
Comment 6: 3. The PSO algorithm is a classical algorithm that has been utilized in different scenarios. While the authors mention their suggested version, the improvement to the algorithm is trivial and lack the novelty. Further, it is not clear how the algorithm modifies duty cycle parameters dynamically. This needs to be presented in detail, accompanied by steps and pseudocode of the new version of the algorithm.
Response 6: We appreciate your insights regarding the PSO algorithm and the need for clarity on the dynamic change. In response, we have revised our manuscript to explain the improvements of the PSO-DC algorithm in more detail and emphasise its novelty and practical importance. We have also included a detailed description of the dynamic modification process for the duty cycle parameters, accompanied by the steps and pseudocode of the new version of the algorithm. With these revisions, we aim to clarify the novelty and practical significance of our PSO-DC algorithm and explain in detail how it dynamically modifies the duty cycle parameters in WSNs.
Comment 7: 4. The flowchart of the WSN simulation on page 13 does not provide anything new.
Response 7: We understand your concern that at first glance, the flowchart on page 13 (now on page 10) does not contain any new concepts. However, we have highlighted in detail the unique aspects of the new contributions and improvements that our flowchart is intended to illustrate in our revised version. To address your concerns, we have provided a more detailed explanation in the accompanying text to clarify the unique aspects of our approach and how it differs from traditional methods. We believe that these improvements will better convey the innovative elements of our WSN simulation flowchart and its relevance to IIoT applications.
Comment 8: 5. The authors should focus on detailing their new system instead of including extraneous information.
Response 8: We appreciate your suggestion to focus more on the details of our new system. We recognise that clear and concise information about our proposed system is crucial for readers to recognise its importance and novelty. We have removed superfluous information so that the reader can continue to focus on the core contribution of our research. We ensure that each section of the manuscript focuses on explaining and supporting the proposed system. Background information is summarised succinctly to provide context without distracting from the main narrative. By focusing on these improvements, we aim to provide a clear, detailed and focussed account of our new system that makes it easier for readers to understand its contributions and significance.
Comment 9: 6. It is unclear how the authors set the simulation environment parameters in Table 1. Whether these parameters are randomly selected or based on literature?
Response 9: We appreciate your attention to detail and understand the importance of clarifying the basis for the simulation environment parameters listed in Table 1. The parameters listed in Table 1 were selected based on a thorough review of the relevant literature. We have chosen values that are commonly used in similar studies to ensure that our results are comparable and based on sound research. In addition to the values from the literature, we conducted preliminary experiments to validate and fine-tune these parameters. This empirical validation ensures that the parameters accurately reflect realistic conditions and increase the reliability of our simulation results.
Comment 10: 7. The proposed algorithm should be compared to recent existing ones to prove its novelty and superior performance.
Response 10: We recognise that a thorough comparison is essential to validate the effectiveness and innovation of our work. A detailed comparative analysis of our proposed PSO-DC algorithm with the state-of-the-art algorithms in the field has been mentioned in Section 5 and includes the selection of relevant and widely recognised algorithms from the recent literature and the comparison of their performance with our algorithm.
Comment 11: 8. The reference list is limited and can be further improved.
Response 11: We appreciate your suggestion and have expanded the reference list to include a broader range of relevant and recent studies. This will provide a more comprehensive background and support for our research. We hope that the expanded references will fulfil your expectations and improve the quality of the manuscript.
Comment 12: 9. The authors are recommended to consult additional references on how to discuss and detail their contributions effectively.
Response 12: We have consulted additional references to improve our discussion and detailing of our contributions. These new insights have helped us to improve the clarity and impact of our manuscript. We believe that these improvements will better emphasise the importance of our work.
Comment 13: Comments on the Quality of English Language
The authors have to revise the entire article
Response 13: We thank you for your feedback. We recognise the need for a comprehensive revision and are committed to thoroughly revising and improving the entire article. We will ensure that all sections are improved in terms of clarity, coherence and contribution to the field. We appreciate your input and will endeavour to make the necessary improvements.

Reviewer 2 Report
Comments and Suggestions for Authors
In this paper, the author proposes to integrate 6LoWPAN (IPv6 over low-power wireless personal area network) and message queue telemetry transmission (MQTT-N) in the industrial Internet of Things system, which is used to detect specific events such as abnormal temperature, speed change, overload condition, maintenance alarm and conveyor belt tension in the production line to ensure smooth operation, quality control and safety. The research content is interesting, but there are some problems in the paper:
1. The abstract of the paper does not show the author's emphasis and innovation.
2. Figure 1 is too low to see the key points and new ideas.
3. The first few figures in the paper, Figure 2-4, can't reflect the characteristics of industrial Internet of Things, which is too simple.
4. The current work analysis is not enough and thorough.
5. The simulation environment of WSN simulation is not introduced and described in detail.
6. The figures, data, time delay and other contents compared by the algorithms in Figures 8 and 9 are too simple. It's just a simple analysis of the performance comparison under different ranges, which is too simple.
7. The coverage performance of the network in Figure 13 is also very simple.
8. The summary part is too simple, and the key points are not highlighted. The future work is not clear.
Comments on the Quality of English LanguageIn this paper, the author proposes to integrate 6LoWPAN (IPv6 over low-power wireless personal area network) and message queue telemetry transmission (MQTT-N) in the industrial Internet of Things system, which is used to detect specific events such as abnormal temperature, speed change, overload condition, maintenance alarm and conveyor belt tension in the production line to ensure smooth operation, quality control and safety. The research content is interesting, but there are some problems in the paper:
1. The abstract of the paper does not show the author's emphasis and innovation.
2. Figure 1 is too low to see the key points and new ideas.
3. The first few figures in the paper, Figure 2-4, can't reflect the characteristics of industrial Internet of Things, which is too simple.
4. The current work analysis is not enough and thorough.
5. The simulation environment of WSN simulation is not introduced and described in detail.
6. The figures, data, time delay and other contents compared by the algorithms in Figures 8 and 9 are too simple. It's just a simple analysis of the performance comparison under different ranges, which is too simple.
7. The coverage performance of the network in Figure 13 is also very simple.
8. The summary part is too simple, and the key points are not highlighted. The future work is not clear.
Author Response
Comment 1: (x) Extensive editing of English language required
Response 1: Thank you for your feedback. We have conducted a thorough review of the manuscript and made extensive changes to improve the clarity and accuracy of the English language throughout the manuscript. We hope that these revisions meet your expectations.
Comment 2:
|
|
Must be improved |
|
Does the introduction provide sufficient background and include all relevant references? |
(x) |
|
Is the research design appropriate? |
(x) |
|
Are the methods adequately described? |
(x) |
|
Are the results clearly presented? |
(x) |
|
Are the conclusions supported by the results? |
(x) |
Response 2: Thank you for your constructive feedback. We have made the following improvements to address your concerns:
- We have improved the introduction to provide more comprehensive background information and included all relevant references to provide a comprehensive context for our research.
- We have reviewed and confirmed the appropriateness of our research design and ensured that it aligns well with the aims of the study.
- The methods section has been expanded and clarified to provide a more detailed and transparent description of our procedures.
- We have revised the presentation of our results to improve clarity and ensure that the results are easy to understand.
- We have carefully reviewed our conclusions to ensure that they are fully supported by the results.
We believe that these improvements will enhance the overall quality and clarity of our manuscript.
Comment 3: In this paper, the author proposes to integrate 6LoWPAN (IPv6 over low-power wireless personal area network) and message queue telemetry transmission (MQTT-N) in the industrial Internet of Things system, which is used to detect specific events such as abnormal temperature, speed change, overload condition, maintenance alarm and conveyor belt tension in the production line to ensure smooth operation, quality control and safety. The research content is interesting, but there are some problems in the paper:
Response 3: Thank you for your feedback and for recognising the interesting aspects of our research on the integration of 6LoWPAN and MQTT-SN in industrial IoT systems for anomaly detection. We recognise that there are areas in the paper that need improvement. We are committed to addressing the identified issues to improve the clarity, depth and overall quality of our manuscript. Your constructive comments will inform our revision and we are grateful to you for helping us improve our work.
Comment 4: 1. The abstract of the paper does not show the author's emphasis and innovation.
Response 4: We understand your request to revise the abstract. By revising the abstract to clearly emphasise the key contributions and innovations of our work, we aim to provide a more compelling and informative overview of our research. This improved summary will better convey the significance and implications of our findings and address the concerns raised.
Comment 5: 2. Figure 1 is too low to see the key points and new ideas.
Response 5: We have revised the font of Figure 1 to improve its clarity and readability and to ensure that key points and new ideas are briefly mentioned in the manuscript. We hope that this improvement addresses your concerns and that the figure better conveys the intended information.
Comment 6: 3. The first few figures in the paper, Figure 2-4, can't reflect the characteristics of industrial Internet of Things, which is too simple.
Response 6: We appreciate your insight that our illustrations need to better reflect the characteristics of the Industrial Internet of Things (IIoT). In response to your comment, we have revised Figures 2-4 to better represent the complexity and characteristics of IIoT systems.
- We keep Figure 2 as simple as possible so that readers can understand the working principle of the chosen protocol. The characteristics of IIoT are explained in detail in the written text of the revised manuscript. The fonts in Fig. 2. have been enlarged for better readability.
- 3. has been improved and shows the characteristics of IIoT.
- 4 has been moved to Fig. 5 and we have redrawn the simulation of node placement in the simulator to increase clarity as it represents the detailed setup of the WSN.
The order of the figures is now different as we have revised the whole article according to the reviewers' comments.
Comment 7: 4. The current work analysis is not enough and thorough.
Response 7: We sympathise with the thorough review and understand the need for a more comprehensive and detailed analysis to improve the robustness and clarity of our work. We hope that the revised manuscript focuses on the analysis of the current work and justifies our proposed model.
Comment 8: 5. The simulation environment of WSN simulation is not introduced and described in detail.
Response 8: We recognise the importance of a detailed description of the simulation environment to ensure the reproducibility and clarity of our work. We assure that the revised version with these improvements and the extended analysis will not only confirm the effectiveness of our proposed algorithm and protocols, but also demonstrate its practical applicability and superiority over existing solutions.
Comment 9: 6. The figures, data, time delay and other contents compared by the algorithms in Figures 8 and 9 are too simple. It's just a simple analysis of the performance comparison under different ranges, which is too simple.
Response 9: Thank you for your feedback. We understand that the numbers, data and analyses in Figures 8 and 9 may appear too simplistic. To address this, we will provide a more detailed and comprehensive analysis, including additional metrics and a deeper discussion of the performance comparisons under different scenarios. Our revised manuscript demonstrates the robustness of data collection and the impact of our proposed algorithm compared to existing solutions. Your contribution is valuable and we appreciate your comments to improve our manuscript.
Comment 10: 7. The coverage performance of the network in Figure 13 is also very simple.
Response 10: We recognise the importance of a more comprehensive and detailed analysis of the network coverage to illustrate the effectiveness of our proposed algorithm. In the revised version, Fig. 13 has been moved to Fig. 7. a) shows the schematic representation of the network topology and b) shows the topology of the network in the simulator. We used the simple network topology to analyse the performance of the sensor nodes modified with SSAIL for industrial applications. However, to extend this mapping with functional details, data flow and environmental factors, an interactive version of the diagram is needed to show how the network changes over time, including the mobility of the nodes and the different connection quality, which we envisage for our future work.
Comment 11: 8. The summary part is too simple, and the key points are not highlighted. The future work is not clear.
Response 11: We thank you for your comments and have made the following changes to address your concerns. We have expanded the summary to highlight the key points and findings of our research and ensure that the main contributions are clearly stated. The summary now includes a more detailed discussion of the critical aspects of our work and highlights the improvements in energy efficiency, network performance and data throughput that we have achieved through our optimisation efforts. We have outlined specific directions for future research, including further improvements to power management strategies, exploring advanced routing protocols, and applying our optimisation techniques in various real-world scenarios to validate and extend our results. Future work is mentioned in a separate section 6. We hope that these changes will address your concerns and improve the clarity and impact of our manuscript.
Comment 12: Comments on the Quality of English Language
Response 12: We thank you for your feedback. We believe that all of the above Comments need to be reiterated in order to improve the quality of the English language. We recognise the need for a comprehensive revision and are committed to thoroughly revising and improving the entire article. We will ensure that all sections are improved in terms of clarity, coherence and contribution to the field. We thank you for your input and will endeavour to make the necessary improvements.

Reviewer 3 Report
Comments and Suggestions for Authors
In this study the authors are proposing, integration of 6LoWPAN (IPv6 over Low-Power Wireless Personal Area Networks) and Message Queuing Telemetry Transport for Sensor Networks (MQTT-SN) in an industrial IoT system in order to detect specific events of anomalies temperature, speed variations, overload conditions, maintenance alerts and conveyor belt tensions in the production lines for ensuring smooth operation, quality control and safety.
The manuscript is clearly written, comprehensive, and well-organized as a scientific paper. Figures and captions are clear and appropriately organized. The introduction is clearly written, and the focus of this study is unique. The methodology followed is descripted in full details. The advancements in the technologies & protocols involved are presented and well cited. The importance of the study is understandable.
Unless I am mistaken Figure 1, Figure 2 and Table II are not referenced in the text.
Author Response
Comment 1: (x) I am not qualified to assess the quality of English in this paper
Response 1: Thank you for your feedback. We have conducted a thorough review of the manuscript and made extensive changes to improve the clarity and accuracy of the English language throughout the manuscript.
Comment 2:
|
|
Yes |
|
Does the introduction provide sufficient background and include all relevant references? |
(x) |
|
Is the research design appropriate? |
(x) |
|
Are the methods adequately described? |
(x) |
|
Are the results clearly presented? |
(x) |
|
Are the conclusions supported by the results? |
(x) |
Response 2: Thank you for your constructive feedback. We have made the following improvements according to the other reviewers:
- We have improved the introduction to provide more comprehensive background information and included all relevant references to provide a comprehensive context for our research.
- We have reviewed and confirmed the appropriateness of our research design and ensured that it aligns well with the aims of the study.
- The methods section has been expanded and clarified to provide a more detailed and transparent description of our procedures.
- We have revised the presentation of our results to improve clarity and ensure that the results are easy to understand.
- We have carefully reviewed our conclusions to ensure that they are fully supported by the results.
We believe that these improvements will enhance the overall quality and clarity of our manuscript.
Comment 3: In this study the authors are proposing, integration of 6LoWPAN (IPv6 over Low-Power Wireless Personal Area Networks) and Message Queuing Telemetry Transport for Sensor Networks (MQTT-SN) in an industrial IoT system in order to detect specific events of anomalies temperature, speed variations, overload conditions, maintenance alerts and conveyor belt tensions in the production lines for ensuring smooth operation, quality control and safety.
The manuscript is clearly written, comprehensive, and well-organized as a scientific paper. Figures and captions are clear and appropriately organized. The introduction is clearly written, and the focus of this study is unique. The methodology followed is descripted in full details. The advancements in the technologies & protocols involved are presented and well cited. The importance of the study is understandable.
Unless I am mistaken Figure 1, Figure 2 and Table II are not referenced in the text.
Response 3: Thank you for your positive feedback and for highlighting the strengths of our manuscript. We appreciate your acknowledgement of the clarity, scope and organisation of our article, as well as the detailed methodology and advancements in the technologies and protocols we discussed.
We apologise for the oversight of not referencing Figure 1, Figure 2 and Table II in the text. Your feedback is invaluable and we will make the necessary corrections to improve the manuscript. Thank you again for your thorough review. The revised manuscript ensures that all figures and tables are properly referenced and integrated into the discussion.

Round 2
Reviewer 1 Report
Comments and Suggestions for Authors
The authors have addressed all my comments
Reviewer 2 Report
Comments and Suggestions for Authors
The author has made detailed revisions according to the reviewer's comments and is ready to accept and publish.
Comments on the Quality of English LanguageThe author has made detailed revisions according to the reviewer's comments and is ready to accept and publish.